# Abscisic Acid and Chitosan Modulate Polyphenol Metabolism and Berry Qualities in the Domestic White-Colored Cultivar Savvatiano

**DOI:** 10.3390/plants11131648

**Published:** 2022-06-22

**Authors:** Dimitrios Evangelos Miliordos, Anastasios Alatzas, Nikolaos Kontoudakis, Angeliki Kouki, Marianne Unlubayir, Marin-Pierre Gémin, Alexandros Tako, Polydefkis Hatzopoulos, Arnaud Lanoue, Yorgos Kotseridis

**Affiliations:** 1Laboratory of Oenology and Alcoholic Beverage Drinks, Department of Food Science and Human Nutrition, Agricultural University of Athens, 75 Iera Odos, 11855 Athens, Greece; nickkont@yahoo.gr (N.K.); akouki27@gmail.com (A.K.); ykotseridis@aua.gr (Y.K.); 2Molecular Biology Laboratory, Department of Biotechnology, Agricultural University of Athens, 75 Iera Odos, 11855 Athens, Greece; aalatzas@aua.gr (A.A.); alexandrosgtako@gmail.com (A.T.); phat@aua.gr (P.H.); 3EA 2106 «Biomolécules et Biotechnologie Végétales», UFR des Sciences Pharmaceutiques, Université de Tours, 31 Av. Monge, F37200 Tours, France; marianne.unlubayir@univ-tours.fr (M.U.); marin-pierre.gemin@univ-tours.fr (M.-P.G.); 4Department of Agricultural Biotechnology and Oenology, International Hellenic University, 1st km Drama-Mikrochori, 66100 Drama, Greece

**Keywords:** *Vitis vinifera* L., Savvatiano, biostimulants, abscisic acid, chitosan, polyphenolic profile, gene expression, phenylpropanoid biosynthesis

## Abstract

During the last decade, several studies demonstrated the effect of biostimulants on the transcriptional and metabolic profile of grape berries, suggesting their application as a useful viticultural practice to improve grape and wine quality. Herein, we investigated the impact of two biostimulants—abscisic acid (0.04% *w*/*v* and 0.08% *w*/*v*) and chitosan (0.3% *w*/*v* and 0.6% *w*/*v*)—on the polyphenol metabolism of the Greek grapevine cultivar, Savvatiano, in order to determine the impact of biostimulants’ application in the concentration of phenolic compounds. The applications were performed at the veraison stage and the impact on yield, berry quality traits, metabolome and gene expression was examined at three phenological stages (veraison, middle veraison and harvest) during the 2019 and 2020 vintages. Results showed that anthocyanins increased during veraison after treatment with chitosan and abscisic acid. Additionally, stilbenoids were recorded in higher amount following the chitosan and abscisic acid treatments at harvest. Both of the abscisic acid and chitosan applications induced the expression of genes involved in stilbenoids and anthocyanin biosynthesis and resulted in increased accumulation, regardless of the vintage. Alterations in other phenylpropanoid gene expression profiles and phenolic compound concentrations were observed as well. Nevertheless, they were mostly restricted to the first vintage. Therefore, the application of abscisic acid and chitosan on the Greek cultivar Savvatiano showed promising results to induce stilbenoid metabolism and potentially increase grape defense and quality traits.

## 1. Introduction

Viticulture and winemaking are an indispensable part of the Greek national culture for over 4000 years. An area of more than 60,000 ha is covered with wine grapes, consisting of more than 90% domestic cultivars, namely Savvatiano, Assyrtiko (white-colored), Roditis (white- or pink-colored), Agiorgitiko and Xinomavro (red-colored). In particular, Savvatiano is the most widely planted domestic cultivar, covering an area of 10,370 ha and representing more than 16% of the Greek vineyard [1].

The variation in topography and climate (i.e., mountainous and arid regions) combined with the depopulation of rural areas, have led to the particularity of the Greek vineyard, which consists mainly of small, traditional, family-owned plots. During the last decade the advances in the oeno-viticultural field have focused on the quality of the final product by linking innovative techniques to the unique “terroir” characteristics, in order to achieve the goal of sustainability. The extended summer drought and higher temperatures expected in the Mediterranean area indisputably will have a negative impact on grape production and quality [2]. Therefore, the development and application of sustainable practices are of great importance for Mediterranean viticulture.

So far, several research projects have been conducted, in order to establish a sustainable viticulture [3] ensuring grape quality, especially in the context of the climate change scenario [4,5]. Among the several viticultural techniques developed, the use of biostimulants represents a significant solution. Biostimulants are biomolecules of different origins, such as oligo-polysaccharides, peptides, proteins, lipids and plant hormones, that constitute alternative solutions for pest and disease management [6,7,8,9]. However, benzothiadiazole, chitosan, methyl jasmonate and abscisic acid are also known to improve grape berry quality traits [10,11,12]. These compounds induce various physiological responses in plants, including the activation of secondary metabolism pathways and antioxidant mechanisms, by modulating the corresponding genes’ expression [6,13,14,15,16].

Advances in the winemaking methods have led to an awareness of phenolic compounds’ impact on wine composition [17]. Phenolic composition is a crucial quality parameter in wines, mainly in red wines, contributing to their organoleptic characteristics, such as color, flavor, texture and astringency, and their antioxidant properties [18]. In the case of white wines, phenolic compounds are related with oxidation, and mainly the enzymatic oxidation of the must [19]. Moreover, stilbenes are associated with human health benefits and possess many anti-inflammatory [20], antioxidant [21], cardioprotective [22], anti-diabetic [23], anticancer [24] and anti-aging properties [25]. The wine phenolic attributes are determined by the grape composition at harvest [26], as well as the winemaking practices applied [27]. For instance, the fermentation temperature, maceration time, yeast selection and skin-juice mixing techniques are known to modulate the phenolic profile of the produced wine. Despite the improvement in winemaking techniques, the quality of the produced wine is related directly with the grapes’ quality [28]. Therefore, winemakers and researchers have focused on optimizing the grape quality

In the last two decades, it has been demonstrated that the application of biostimulants can increase the accumulation of phenolic compounds, such as anthocyanins and stilbenes [29,30,31,32,33] as well as the aromatic compounds [10,34] in wine grapes. In particular, abscisic acid and chitosan applications were found to upregulate the phenylpropanoid pathway genes that are critical in phenolic biosynthesis [13,15,35,36]. Abscisic acid is a well-known plant hormone participating in many developmental processes, such as ripening and drought response [12,37]. Several research studies have shown that the exogenous application of ABA on the grapevine improves berry color, due to the accumulation of anthocyanins [35,38,39]. On the other hand, chitosan is a polysaccharide known to induce plant defense and antioxidant mechanisms [16], as well as to improve the aromatic profile of the grape berries [10,11,40]. However, the effect of the treatment depends on the cultivar [41], the dose of the biostimulant [42,43], the time of application [30,42,43,44] and the environmental conditions [45].

Although several studies investigated the effect of biostimulants on the phenolic composition of the red grape and wines, only a few studies focused on the white grape varieties [46,47,48] The current study is the first attempt to assess the influence of biostimulants’ application on grape-berry secondary metabolism of a white grape variety under the Greek vineyard conditions. A thorough molecular and biochemical analysis was performed in parallel to the berry quality traits, in order to elucidate the effects of abscisic acid and chitosan applications on the phenolic composition in the leading white-colored cultivar, Savvatiano. Further, we investigated the biostimulants’ effect on the expression profile of genes encoding key enzymes in the phenylpropanoid pathway.

## 2. Results

### 2.1. Climate Conditions

The experimental vineyard is located in a narrow valley at approximately 450 m altitude, in the area of Central Greece, 100 Km north-west of Athens (Appendix A). The Mediterranean climate of the Muses Valley is characterized by mild, wet winters and dry summers. Based on the meteorological data recorded at the nearby town of Askri, Viotia, monthly maximum temperatures were higher during the summer months (June- August) of the 2019 vintage (39.5 °C) compared to 2020 (37 °C) (Figure 1A). In contrast, the total amount of rainfall in the same period was higher in 2020 (96.5 mm) compared to 2019 (37 mm) (Figure 1A). Remarkably, during the first two weeks of September (i.e., the onset of the veraison to middle veraison stage) more days with a temperature of over 30 °C were recorded in 2020 than the same period of the 2019 vintage (Figure 1B). As a consequence, the ripening period (i.e., veraison to harvest) of the 2020 vintage was shorter, due to the higher maximum temperatures.

### 2.2. Berry Ripening and Conventional Must Analysis

#### 2.2.1. Berry Size

During the 2020 vintage, the berry size in both the control and biostimulant-treated vines increased significantly in all of the samples throughout ripening and at the harvest stage (Table 1), compared to 2019. As mentioned above, the total amount of rainfall of July and August was higher in 2020 compared to the previous year (Figure 1). During this period, the grape berries gather a significant amount of water which increases the berry weight, which could explain the differences observed between the two vintages. The results from the biostimulant treated vines showed that during the 2019 vintage the highest concentrations applied had a greater effect on grape berry weight. Specifically, the average weight of berries sprayed with the highest concentrations of ABA and CHT (2.08 g and 2.05 g at harvest, respectively) was significantly lower (*p* < 0.05) than the control (2.30 g) and berries treated with the low concentrations (2.55 g) at the harvest stage (Table 1). Significant differences between the treatments were observed in the berry size during the second vintage at the harvest, while during the ripening period no differences were observed. In contrast to the 2019 vintage, in the 2020 vintage the highest doses of ABA and CHT exhibited significantly increased berry weight than the control at the harvest stage (Table 1). The higher levels of weight/ berry in the 2020 compared to the 2019 vintage could be linked to the higher rainfall.

#### 2.2.2. Total Soluble Solids

Among the most important berry quality traits is the grape juice composition, determined mainly by the total soluble solids (TSS; expressed in °Brix) and the titratable acidity (TA; expressed in tartaric acid g/L). The grape berries in both of the vintages exhibited a constant increase in the total soluble solids during maturation (Table 1). The TSS concentration in the control vines in almost all of the sampling dates of the 2019 vintage was lower than in the 2020 vintage, implying that the high temperatures observed during July and August of 2019 (Figure 1A) could have a negative impact on the subsequent sugar accumulation (Table 1). Biostimulant application induced variations in the TSS concentration in both of the vintages. The highest concentrations of ABA and CHT resulted in higher °Brix at the harvest stage in the 2019 vintage than the untreated and lowest concentrations, while no significant differences were observed between the controls and the berries treated with the lower elicitor concentrations (Table 1). However, during the 2019 ripening period no differences were recorded between the treatments and the control berries. On the other hand, the ABA and CHT treatments during the 2020 vintage resulted in decreased TSS concentrations, compared to the control in all of the sampling dates (Table 1), indicating that the elicitors’ application, together with the high temperatures of September 2020 (Figure 1), had a negative impact on TSS.

#### 2.2.3. pH and Total Acidity

Statistically significant differences in the titratable acidity were observed between the controls and treated berries at the first two phenological stages of both of the vintages (Table 1), with the highest CHT dose providing a higher value of TA in the veraison and middle veraison sampling dates. In contrast, both the ABA and CHT treated berries exhibited a lower level of TA, compared to the control vines, at the harvest stage in 2019 (Table 1), while the TA level in both of the treatments was higher than the controls at the harvest stage in the following vintage (Table 1). Similarly to TA, differences in the pH levels were observed during the ripening period in both of the vintages, while the variations between the treated and control berries at the harvest stage concurred with the TA results. Specifically, the pH level was increased in the treated berries, with the highest dose of ABA or CHT, at the harvest of 2019, and decreased at the harvest of 2020, compared to the untreated ones (control) (Table 1).

### 2.3. Metabolic Changes in Grape Berries

The grape berries were collected from the control and treated vines at three phenological stages (veraison, middle veraison and harvest) and a metabolic profiling was performed by UPLC–MS. A total of 30 metabolites were detected and identified by UPLC–MS in the berry samples (Table 2), comprising of six amino acids, four phenolic acids, four stilbenoids, six flavonols, eight flavan-3-ols and two anthocyanins di-OH.

#### 2.3.1. Multivariate Statistics

To investigate the abscisic acid-induced changes on the berry metabolism, different subsets of the metabolomic data at veraison, middle veraison and harvest were analyzed by unsupervised analysis. The PCA score plot of the berries at veraison from 2019 and 2020 explained 50.5% of the variance and showed that the vintage effect was the main factor of discrimination, with a separation along the PC1 axis (Figure 2A). Additionally, the samples were separated along PC2, showing the impact of the ABA treatments on berry metabolism. The corresponding loading plot showed the metabolites responsible for this separation (Figure 2B). As a result, the metabolites were mainly clustered in relation to their structural class, especially flavan-3-ols. The projection of the flavan-3-ols on the PC2 negative axis showed that these compounds were responsible for the discrimination between the control and treated samples, with a decrease in the flavan-3-ols in the ABA-treated berries. Similar PCA analyses were performed to explore the chitosan-induced metabolic changes in the berries. The PCA score plot of the berries at veraison from 2019 and 2020 explained 54.8% of the variance, with a clear separation between the control and treated samples, whereas no vintage effect was observed (Figure 2C). In this experiment, the loading plot also showed that the metabolites were highly clustered in relation to their structural class, with a decrease in all of the flavan-3-ols in the chitosan-treated berries (Figure 2D).

The PCA score plots of the berries at the middle veraison and harvest stages showed no discrimination between the control and treated samples, suggesting that ABA temporarily impacted the berry metabolism at the veraison stage, with no visible metabolic changes at the middle veraison and harvest stages (Appendix A). As well as for ABA treatments, the PCA score plots at middle veraison and harvest showed no discrimination between the control and CHT-treated berries (Appendix A).

#### 2.3.2. Univariate Statistics

The application of the biostimulants did not affect the total amino acid level at any of the three stages during the vintage of 2019. However, the ABA and CHT applications resulted in significantly higher concentrations of total amino acids at the middle veraison stage of 2020 (Figure 3A; Appendix A).

The concentration of the anthocyanins was found to be constant during the ripening of the control vines. All of the biostimulant treatments led to increased total anthocyanin levels at the veraison and middle veraison stages during the vintage 2019 (Figure 3B; Appendix A). However, the positive effect on the anthocyanin concentration at the harvest stage was observed only in the vines treated with the highest doses of ABA and CHT (Appendix A). On the other hand, increased anthocyanins were observed during the 2020 vintage only at the veraison stage in the samples from the vines treated with the highest dose of ABA (Figure 3B; Appendix A).

The total flavan-3-ols were reduced during maturation in both of the vintages, but no differences were observed after the biostimulants’ applications during the vintage of 2019 (Figure 3C; Appendix A). On the contrary, the treatment effects were evident in the following year. At the veraison stage of 2020, the flavan-3-ols were negatively affected by the biostimulant treatments, especially by the chitosan (Figure 3C). Interestingly, the highest CHT dose increased the flavan-3-ols at the middle veraison stage (Appendix A). However, no differences were observed at the harvest stage during the 2020 vintage (Appendix A).

The phenolic acids represent an important group of compounds, with a key role in the white varieties. In this study, four phenolic acids were detected by UPLC–MS, namely gallic acid, coutaric acid, caftaric acid and fertaric acid. A consistent decrease in the total phenolic acids’ concentration in the grape berries was recorded over the ripening period, while the ABA and CHT treatments did not lead to any significant change (Figure 3D; Appendix A). It is worth mentioning that the biostimulant applications resulted in a reduced total phenols level compared to the controls in the 2019 vintage, while the opposite effect was observed in the following year. However, these differences were not statistically significant (Figure 3D; Appendix A).

The flavonols were decreased towards the harvest stage in the control plants in both of the vintages (Figure 3E; Appendix A). Both of the biostimulant applications induced flavonols at the veraison stage of the 2019 vintage. The flavonol concentrations were slightly lower in the ABA- and CHT-treated vines (Figure 3E). At the other two phenological stages, no differences between the treated and control vines were recorded in both of the vintages, except for the significantly lower concentration measured at the harvest of 2020 in those grapes treated with the higher dose of CHT (Appendix A).

Another important group of phenolic compounds detected by the UPLC–MS was the stilbenoids, namely piceid, *E*-resveratrol, *E*-piceatannol and *E*-ε-viniferin. The concentration of stilbenoids in the control berries was increased during ripening, peaking at the middle veraison stage (Figure 3F; Appendix A). Although no significant differences were observed between the treated and untreated vines at the veraison stage of 2019, the highest ABA treatment showed a significantly higher (i.e., two to three-fold) concentration of stilbenoids, compared to the controls and the other treatments, at the same phenological stage of 2020 (Figure 3F). The effect of the ABA and CHT application was negligible at the middle veraison stage of both of the vintages (Appendix A), while it was evident at the harvest stage. Remarkably, the lowest dose of the ABA treatment resulted in a significantly increased total stilbenoids level in both of the vintages, while the higher dose CHT treatment exhibited a similar effect in the 2019 vintage (Appendix A).

Taken together, the UPLC–MS results indicate that most of the changes in the phenolic composition due to the different biostimulant applications were observed during the veraison stage in both of the vintages. In general, the ABA and CHT treatments resulted in higher stilbenoid and anthocyanin levels and, on the other hand, in a decreased flavan-3-ols concentration. However, the effect of the treatments on the phenolic compounds’ concentrations was found to be mostly dependent on the dose and the vintage.

### 2.4. Gene Expression Analysis

Recent results have highlighted that biostimulants cause alterations in the grape berry transcriptome [13,14,15,49,50]. In the present study, the effect of biostimulant applications on the gene expression was examined by targeted RT-qPCR analysis of the berry samples collected at three different stages (veraison, middle veraison and harvest) during the 2019 and 2020 vintages.

We initially investigated the expression profile of the phenylalanine ammonia lyase (*VviPAL*) and cinnamate 4-hydrolase (*VviC4H*) genes that encode the first two enzymes of the phenylpropanoid pathway [51]. The expression analysis of the *VviPAL* gene in the control vines showed an upregulation after veraison, followed by a slight decrease at the harvest stage during the vintage of 2019. However, the transcript level remained constant throughout berry maturation in 2020 (Figure 4; Appendix A). The ABA applications altered the expression profile of *VviPAL* in a dose-dependent manner. A significant upregulation of the *VviPAL* gene at the veraison and harvest stages was observed in the vines treated with the lowest dose of ABA, while a reduced expression level at harvest was observed in the vines treated with the highest dose. The highest dose of CHT reduced the *VviPAL* expression at the beginning of ripening during the first vintage, while both of the chitosan treatments exhibited the same effect at the harvest stage of 2020 (Figure 4; Appendix A). The *VviC4H* transcript level in the control vines was increased after veraison in the 2019 vintage, but remained constant during maturation in 2020 (Figure 4; Appendix A). The lowest dose of ABA resulted in the upregulation of *VviC4H* at veraison of the first vintage, followed by a reduced expression level towards harvest. The highest dose of ABA led to a reduced *VviC4H* expression at harvest of both of the vintages. The lowest dose of CHT resulted in the upregulation of the *VviC4H* gene at middle veraison of 2019, while both of the doses led to a reduced expression level at the harvest stage of both of the vintages (Figure 4; Appendix A).

The stilbene synthase gene (*VviSTS*), initiating the stilbenes’ biosynthesis [52], exhibited a dramatical upregulation throughout ripening, mainly in the first vintage. During the following year, the increase was also evident, but was observed only at the harvest stage (Figure 4; Appendix A). All of the biostimulant treatments in the 2019 vintage resulted in increased *VviSTS* transcript accumulation at the first phenological stages. Furthermore, the application of lower doses of ABA and CHT had the same effect on the expression level in the 2020 vintage (Figure 4; Appendix A).

We further examined the expression patterns of the genes belonging to the biosynthetic pathway of flavonoids. The expression of the flavonol synthase gene (*VviFLS*) that encodes for the enzyme catalyzing the flavonol biosynthesis [53] showed a declining trend during berry maturation (Figure 4; Appendix A). The same trend—and also a lower expression level—was observed in most of the biostimulant treatments, except for the lower doses of ABA and CHT that led to *VviFLS* upregulation at the veraison stage of 2019 and 2020, respectively (Figure 4; Appendix A). The expression of the UDP-glucose-flavonoid 3-O-glycosyltransferase gene (*VviUFGT*), encoding for the critical step in anthocyanin biosynthesis [54], was constant throughout maturation in the 2019 vintage, while it was gradually increased at the harvest stage in the following year (Figure 4; Appendix A). The lowest dose of ABA resulted in the upregulation of the *VviUFGT* gene at middle veraison of 2019, while both of the doses led to a reduced transcript level at the harvest of 2020. On the other hand, the CHT treatments resulted in an increased *VviUFGT* expression at all of the phenological stages, regardless of the vintage (Figure 4; Appendix A). Finally, the expression level of the leucoanthocyanidin reductase 1 gene (*VviLAR1*), involved in the flavan-3-ols’ biosynthesis [55], was decreased during berry maturation, especially in the 2019 vintage (Figure 4; Appendix A). The lowest doses of ABA and CHT resulted in the upregulation of the *VviLAR1* gene at the veraison and middle veraison stages, respectively, during the first vintage. On the other hand, the highest doses of the biostimulants positively influenced the *VviLAR1* expression at middle veraison in 2020 (Figure 4; Appendix A).

Altogether, the results of the gene expression analysis suggest that the application of the lowest dose of ABA had a positive effect on *VviPAL* expression and the application of chitosan similarly influenced the expression of *VviUFGT* in both of the vintages. Moreover, both of the doses of ABA and CHT resulted in an early upregulation of *VviSTS,* while their lower doses affected positively the expression of *VviFLS* (both vintages) and *VviC4H* (in the 2019 vintage).

## 3. Discussion

Muses Valley is characterized by extinguishing topographical and meteorological features that have favored grapevine cultivation since the ancient times. Although the grapevine is considered to be a drought-resilient species traditionally covering semiarid areas, the environmental factors, such as the climatic variability, influence grape and wine quality. The different weather conditions between the two years at the location of the vineyard influenced the maturation process; therefore, the vintage effect was evident in the grape berries’ physiochemical characteristics during the harvest stage. For instance, the higher amount of rainfall during the summer months of 2020 resulted in an increased berry size compared to the previous vintage. Furthermore, differences in berry weight were observed between the treatments, as well as the treated vines that received the highest doses producing significantly larger grape berries. On the other hand, the treatment effect on the berry size was evident under the moderate weather conditions of 2019; hence, the highest dose of biostimulants decreased the berry weight. Vintage-dependent variations were also observed in the grape juice components. However, whilst biostimulant application was found to have a limited effect on the berry sugar content in other studies [38,56], we observed that the highest doses of ABA and CHT resulted in an increased TSS level in 2019, while both of the doses of biostimulants caused a decreased TSS level at the harvest of 2020. On the contrary, berries from the treated vines exhibited a lower TA level than those from the untreated vines at the harvest stage of 2019, while the opposite effect was observed in the following vintage.

It is well-established that biostimulants can increase the deposit of phenolic compounds [29,30,31,32]. However, the outcome of the application is dependent on the environmental conditions [45]. Herein, we showed that the ABA and CHT treatments increased the stilbenoid and anthocyanin concentrations, regardless of the vintage. Although anthocyanins are known to determine the red color of grapes and wines, recent studies demonstrated the existence of small amounts in the white grape berries [57] and wines [58]. The white color of grapes was initially explained by mutations in the VviMYBA1 and VviMYBA2 genes, which regulate the *VviUFGT* expression [59]. However, the existence of several other MYB-type transcription factors that can modulate flavonoid biosynthesis [60], and the identification of *VviUFGT* in transcriptomic studies in white-colored cultivars [51,61], imply more complex regulatory mechanisms.

In contrast to the effect of the stilbenoids and anthocyanins, the positive effect on the other phenolic compounds (i.e., flavonols) was restricted to the first vintage. The increased berry size, observed during the second vintage, possibly led to the dilution of the phenolic compounds, so that the differences between the treatments could not be distinguished. Furthermore, the increased daily maximum temperatures observed during the ripening period of 2020 could have a negative impact on the phenolic compounds. It is known that high temperatures decrease the accumulation of the anthocyanin, phenolic and volatile compounds, and it has been suggested that specific secondary metabolites (e.g., kaempferol and flavonols in general) could be indicators of fruit quality losses associated with warming [62]. The beneficial effect of exogenously applied biostimulants on grape composition can be achieved during moderate climate conditions [63], while the effect is limited under a hot and dry climate [56]—similar to the local weather conditions observed at the experimental vineyard during the ripening period of the second vintage.

Although Savvatiano is a white grape variety, this is the first time that small amounts of anthocyanins were detected in all of the treatments. Although still controversial, it is believed that the presence of these anthocyanins in white wines could provoke the pinking phenomenon [64].

Principal component analyses showed that the biostimulant applications had a strong impact on the grape berry composition at veraison, while no discrimination between the control and treated samples was observed in the following phenological stages. Veraison is the crucial stage in grape berry development. The initiation of berry softening and sugar accumulation, together with the transcriptional reprogramming and alterations in secondary metabolites’ biosynthesis, will lead to the exquisite characteristics of any distinct grapevine cultivar. During recent years, several studies have shown that the critical time period for biostimulant application is the week before or after veraison [12,15,36,40].

Considering that the grape phenolic composition depends on the transcriptional changes induced by environmental conditions, it was challenging to complement our study with an expression analysis of the target genes encoding the key enzymes of the phenylpropanoid biosynthetic pathway. Recently, accumulated results from transcriptomic studies revealed the differential expression of the phenylpropanoid pathway genes, due to biostimulant applications [12,13,14,15,49]. Herein, we showed that the expression level of *VviPAL* and VviC4H was nearly constant during the grape berry maturation in untreated plants, while the *VviSTS* exhibited a significant increase throughout ripening and the *VviFLS* expression was found to decline towards harvest. An increase was also observed in the *VviUFGT* gene expression at the harvest stage of 2020. The ABA and CHT applications resulted in increased *VviPAL* and *VviUFGT* expression levels, respectively, while both of the treatments resulted in an early upregulation of the *VviSTS* gene expression. However, in certain cases the treatment effect on the gene expression was also dependent on the vintage. For instance, the *VviC4H* and *VviFLS* and expression level were increased in the treated plants compared to the controls at the veraison stage of the 2019 vintage, while no difference was observed the following year.

Remarkably, the *VviUFGT* and *VviSTS* expression was positively affected by the biostimulants’ application in both of the vintages, which concurs with the increased concentration of anthocyanins and stilbenoids. Our results indicate that the early upregulation of these two genes at the veraison stage led to higher levels of the corresponding compounds at harvest, contributing to an improved grape berry quality. The positive effect of ABA and CHT on anthocyanin biosynthesis and accumulation has been reported in many studies [13,15,35,38,65] and similar results have been obtained with stilbenoids. More specifically, the treatment of grape berries with ABA [66] and CHT [67], as well as the elicitation of cell suspension cultures [68,69,70], resulted in the upregulation of stilbene synthase genes, accumulation of the encoded enzymes and a significant increase in the stilbenoid content [71]. Considering the important role of stilbenoids in plants’ defense mechanisms, and their numerous benefits to human health, these findings can be useful in future applications.

## 4. Materials and Methods

### 4.1. Experimental Design

The experiment was conducted during two growing seasons (2019 and 2020) in a commercial vineyard in Muses Valley (Askri, Viotia; 38°19′30′′ N, 23°05′37′′ E, at an elevation of 450 m) in Central Greece (Appendix A), planted with the *Vitis vinifera* L. Savvatiano. The vines were more than 50 years old and pruned as bush vines. The vineyard was located on a deep loamy soil and was managed according to standard agronomical practices of the region, without irrigation. The number and timing of the viticultural practices (i.e., plant protective applications) were similar for all of the treatments. The experiment was conducted in a randomized block design, with all of the treatments applied in three replicates, using 10 vines for each replication (Appendix A). More than three grapevine plants in each row between the different treatments were considered as the buffer parts of the vineyard and were not sampled. The vines were sprayed with an aqueous solutions of 0.04% *w*/*v* (low dose) or 0.08% *w*/*v* (high dose) abscisic acid (s-abscisic acid 10.4% *w*/*v*, Protone SL, Hellafarm, Peania, Greece; ABA treatment) and 0.3% *w*/*v* (low dose) or 0.6% *w*/*v* (high dose) chitosan (chitosan hydrochloride 3% *w/w*, Project One, Phytorgan S.A., Kifisia, Greece; CHT treatment). Aquascope (Hellafarm, Greece) and Tween 80 (Sigma-Aldrich, St. Louis, MI, USA) were used as wetting agents for the ABA and CHT treatments, respectively. None of the sprayed vines served as the control (control treatment). For the ABA treatment, spraying was performed at the grape zone (clusters only) at the veraison stage (81–85, according to the BBCH scale) [72], and a second and a third application were performed 3 and 6 days after the first application (Appendix A). The CHT applications were carried out on the whole vine canopy (leaves and clusters) at the veraison stage with a second and a third application performed 7 and 14 days after the first application (Appendix A).

### 4.2. Sampling and Physicochemical Determination in Grapes

The grapes were harvested on 29 September 2019 and 24 September 2020 (Appendix A). In order to follow the ripening process of the grape berries and define the optimum harvest date, sampling was carried out once every week. At all of the sampling points, 50 berries from each experimental treatment were randomly collected and the berries’ fresh weight was determined. The grape maturity level was monitored weekly by measuring the Total Soluble Solids content (°Brix), Titratable Acidity and pH, according to the official methods from the *Compendium of International Methods of Wine and Must Analysis* [73]. The grape berries were collected at three maturity level points: at veraison, just before the first treatment application; at middle veraison and at the optimum pulp maturity stage. A total of 50 berries of each replicate were collected randomly. The samples were transported on dry ice and were carefully stored at −80 °C until required for analysis.

### 4.3. UPLC–MS-Based Metabolic Profiling

The frozen grape berries (−80 °C) were ground to powder with liquid nitrogen, after the elimination of the seeds, and used for metabolic profiling, based on the adapted methods from previous studies [14,74]. Fifty mg of ground-to-powder berry dry weight was extracted using 1 mL of 80% (*v*/*v*) methanol. After 30 min of sonication, the samples were macerated overnight at 4 °C in the dark and centrifuged at 18,000× *g* for 10 min. The supernatant was diluted five-fold in 80% (*v*/*v*) methanol and stored at −20 °C prior to further analyses. The UPLC–MS was performed using an ACQUITY™ Ultra Performance Liquid Chromatography system coupled to a photo diode array detector (PDA) and a Xevo TQD mass spectrometer (Waters, Milford, MA, USA), equipped with an electrospray ionization (ESI) source controlled by Masslynx 4.1 software (Waters, Milford, MA, USA). The analyte separation was achieved by using a Waters Acquity HSS T3 C18 column (150 × 2.1 mm, 1.8 μm), with a flow rate of 0.4 mL min^−1^ at 55 °C. The injection volume was 5 μL. The mobile phase consisted of solvent A (0.1% formic acid in water) and solvent B (0.1% formic acid in acetonitrile). The chromatographic separation was achieved using an 18-min linear gradient from 5 to 50% solvent B. The MS detection was performed in both the positive and negative modes. The capillary voltage was 3000 V and sample cone voltages were 30 and 50 V. The cone and desolvation gas flow rates were 60 and 800 Lh^−1^. The identification of the analytes was based on retention times, *m*/*z* values and UV spectra, and by comparison with commercial standards, own purified compounds or data from literature when no authentic standards were available. The complete description of analyte identification can be seen in Martins et al. [14] and the present ID numbers are as follows: L-proline (m1); L-leucine (m2); L-isoleucine (m3); L-phenylalanine (m4); L-tyrosine (m5); L-tryptophan (m6); cyanidin-3-*O*-galactoside (m7); peonidin-3-*O*-(6-*p*-coumaroyl-glucoside) (m8); gallic acid (m9); catechin (m10); epicatechin (m11); coutaric acid (m12); caftaric acid (m13); fertaric acid (m14); E-piceid (m15); catechin-gallate (m16); kaempferol-3-*O*-glucoside (m17); quercetin-3-*O*-glucoside (m18); quercetin-*O*-glucuronide (m19); quercetin-3-*O*-glucuronide (m20); myricetin-glucoside (m21); procyanidin B1 (m22); procyanidin B2 (m23); procyanidin B3 (m24); procyanidin B4 (m25); procyanidin-gallate (m26); *E*-resveratrol (m27); *E*-piceatannol (m28); *E*-ε-viniferin (m29); kaempferol-3-*O*-rutinoside (m30) (Table 2) The extraction and UPLC–MS analyses were performed in triplicates.

**Table 2 plants-11-01648-t002:** List of compounds identified in this study based on MS and UV spectra. RT retention time, * tentative assignments based on MS data, UV spectra, elution order available from literature.

Compound No	Compound Assignement	RT (min)	Compound Class	Molecular Ion Adducts ES^+^	In Source Fragment ES^+^	Molecular Ion Adducts ES^−^	In Source Fragment ES^−^	λ_max_ (nm)	References
m1	L-proline	1.01	Amino acid	116 [M+H]^+^				225, 275	Standard
m10	citric acid	1.43	Organic acid			191 [M-H]^−^	173 [M-H_2_O-H]^−^ 111 [173-O=C(OH)_2_]^−^	200	Standard
m5	L-tyrosine	1.51	Amino acid	182 [M+H]^+^				207	Standard
m2	L-leucine	1.61	Amino acid	132 [M+H]^+^	86 [M-CH(CH_3_)_2_-2H+H]^+^ (immonium ion)			201, 232	Standard [75]
m3	L-isoleucine	1.74	Amino acid	132 [M+H]^+^	86 [M-CH(CH_3_)_2_-2H+H]^+^ (immonium ion)			210, 268	Standard [75]
m9	gallic acid	1.94	Phenolic acid			169 [M-H]−	125 [M-CO_2_]−	210, 271	Standard [76]
m4	L-phenylalanine	2.72	Amino acid	166 [M+H]^+^	148 [M-H_2_O+H]^+^ 120 [M-H_2_O-CO+H]^+^			200, 280	Standard
m13	caftaric acid	3.63	Phenolic acid			311 [M-H]^−^	179 [M - tartaric acid]^−^ 149 [tartaric acid - H]^−^ 135 [caffeic acid - COO]^−^	200, 229, 328	Standard [77] [78] [79]
m6	L-tryptophan	3.82	Amino acid	205 [M+H]^+^				219, 269	Standard
m22	procyanidinB1	4.26	Flavan-3-ol	579 [M+H]^+^	427 [M+H-C8H8O3]+ (RDA) 291 [M+H-(epi)catechin]+ (QM)	577 [M-H]^−^	425 [M-H-C_8_H_8_O_3_]^−^ (RDA) 407 [M-H-C_8_H_8_O_3_-H_2_O]^−^ 289 [M-H-(epi)catechin]^−^ (QM)	280, 313	Standard [77] [80]
m24	procyanidinB3	4.58	Flavan-3-ol	579 [M+H]^+^	427 [M+H-C8H8O3]+ (RDA) 291 [M+H-(epi)catechin]+ (QM)	577 [M-H]^−^	425 [M-H-C_8_H_8_O_3_]^−^ (RDA) 407 [M-H-C_8_H_8_O_3_-H_2_O]^−^ 289 [M-H-(epi)catechin]^−^ (QM)	200, 275sh	Standard [77] [80]
m12	coutaric acid	4.66	Phenolic acid			295 [M-H]^−^	163 [coumaric acid - H]− 149 [tartaric acid - H]− 119 [coumaric acid - COO]^−^	205, 311	Standard [81]
m10	catechin	4.77	Flavan-3-ol	291 [M+H]^+^		289 [M-H]^−^	271 [M-H-H_2_O]^−^ 245 [M-H-CO_2_]^−^ 205 [M - A ring]^−^ 203 [M-H-CO_2_-C_2_H_2_O]^−^	229, 278	Standard [82]
m14	fertaric acid	4.98	Phenolic acid			325 [M-H]^−^	193 [ferulic acid - H]^−^ 149 [tartaric acid - H]^−^ 134 [ferulic acid - COO - CH_3_]^−^	221, 262, 340	Standard [77] [78] [83]
m25	procyanidinB4	5.2	Flavan-3-ol	579 [M+H]^+^	427 [M+H-C8H8O3]+ (RDA) 291 [M+H-(epi)catechin]+ (QM)	577 [M-H]^−^	425 [M-H-C_8_H_8_O_3_]^−^ (RDA) 407 [M-H-C_8_H_8_O_3_-H_2_O]^−^ 289 [M-H-(epi)catechin]^−^ (QM)	202, 264, 362sh	Standard [77] [80]
m23	procyanidinB2	5.35	Flavan-3-ol	579 [M+H]^+^	427 [M+H-C8H8O3]+ (RDA) 291 [M+H-(epi)catechin]+ (QM)	577 [M-H]^−^	425 [M-H-C_8_H_8_O_3_]^−^ (RDA) 407 [M-H-C_8_H_8_O_3_-H_2_O]^−^ 289 [M-H-(epi)catechin]^−^ (QM)	200, 278	Standard [77] [80]
m11	epicatechin	5.91	Flavan-3-ol	291 [M+H]^+^		289 [M-H]^−^	271 [M-H-H_2_O]^−^ 245 [M-H-CO_2_]^−^ 205 [M - A ring]^−^ 203 [M-H-CO_2_-C_2_H_2_O]^−^	229, 278	Standard [82]
m26	procyanidin gallate	6.51	Flavan-3-ol	731 [M+H]+	507	729 [M-H]−	505, 523, 577 [M-H-galloyl]−	206, 276	
m22	myricetin-3-*O*-glucoside	7.03	Flavonol			479 [M-H]−	317 [M-H-glucose]	206, 356	[83]
m7	cyanidin-3-*O*-(6-*O*-acetyl)-glucoside	7.16	Anthocyanin diOH	493 [M+H]+	511 [M+H+H2O]+ 287 [M+H-glucose-acethyl]+			202, 264, 325	Standard
m16	catechin gallate	7.73	Flavan-3-ol			441	289, 169, 125	207, 280	[78]
m20	quercetin-3-*O*-glucuronide	8.02	Flavonol	479 [M+H]+	303 [M+H-glucuronic acid]+	477 [M-H]−	301 [M-H-glucuronic acid]− 955 [2M-H]−	256, 359	Standard
m19	quercetin-3-*O*-glucoside	8.14	Flavonol	465 [M+H]+	303 [M+H-glucose]+	477; 463; 478; 941; 955	477; 463; 301; 478; 941; 955	205, 273, 251	Standard
m28	*E*-piceatannol	8.77	stilbenoid DP1	244 [M+H]+		242 [M-H]−		207, 283	[78] & [84]
m30	kaempferol-3-*O*-rutinoside	8.75	Flavonol	595 [M+H]+	449 [M-rhamnose+H]+ 287 [M-rutin+H]+	593 [M-H]− 609 [M+O]+	447 [M-rhamnose-H]− 285 [M-rutin-H]− 301, 271	224, 264, 345	Standard
m15	*E*-piceid	9.42	Stilbenoid glucoside			389 [M-H]−	227, 185	200, 218, 221	Standard
m8	peonidin-3-*O*-(6-*p*-coumaroyl-glucoside))	9.79	Anthocyanin diOH	609 [M+H]+		607 [M-H]−		205, 283	Standard
m27	*E*-resveratrol	11.13	stilbenoid DP1	229 [M+H]+		227 [M-H]−	143, 185	203, 279	Standard
m29	*E*-ε-viniferin	12.53	Stilbenoid DP2	455 [M+H]+		453 [M-H]−	347, 359, 225	225sh, 323	Standard [84]

### 4.4. RNA Extraction and Gene Expression Analysis by RT-qPCR

The grape berries without seeds were ground to powder with liquid nitrogen, and the RNA was extracted by the method of Reid et al. [85]. Briefly, approximately 1 g of ground tissue was extracted with a buffer containing 300 mM Tris-HCl (pH 8.0), 25 mM EDTA, 2 M NaCl, 2% (*w*/*v*) CTAB, 2% (*w*/*v*) PVPP and 0.05% (*w*/*v*) spermine at 65 °C for 15 min, mixed thoroughly with an equal volume of chloroform: isoamyl alcohol (24:1) and centrifuged. The step was repeated for aquatic phase, the RNA was precipitated with 0.6 volumes isopropanol and 0.1 volumes sodium acetate at −20 °C overnight, centrifuged and finally dissolved in 100 μL ddH2O. The RNA samples were treated with DNAse I (Takara Bio, Shiga, Japan) and further purified using phenol: chloroform: isoamyl alcohol (25:24:1), followed by ethanol precipitation. The RNA quantity and quality were determined using a NanoDrop ND-1000 Spectrophotometer (Thermo Fisher Scientific Inc., Wilmington, DE, USA) and verified by 0.8% agarose gel electrophoresis. The reverse transcription was performed with 2 μg RNA using SMART MMLV-Reverse Transcriptase (Takara Bio, Shiga, Japan) and oligo (dT) primer (Eurofins Genomics, Ebersberg, Germany). The synthesized cDNA was five-fold diluted and PCR conditions were optimized for primers corresponding to the selected genes from various metabolic pathways (listed in Table 3). The samples were further diluted and quantitative PCR reactions were performed in the PikoReal Real-Time PCR System (Thermo Fisher Scientific, Vantaa, Finland) using the KAPA SYBR FAST qPCR Master Mix (KAPA Biosystems, Cape Town, South Africa) and applying the following cycler conditions: 2 min at 50 °C, 2 min at 95 °C, followed by 40 cycles of 15 s at 95 °C, 30 s at 62 °C, 30 s at 72 °C. All of the quantitative PCR reactions were performed as triplicates and the melting curve analysis was performed at the end of each reaction to confirm primer specificity. The quantification of gene expression was performed according to the 2^−ΔCt^ method and elongation factor 1a (VviEF1a) was used as the reference gene for data normalization.

### 4.5. Statistical Analysis

All of the values are presented as the mean standard deviation. The statistical analyses were performed using Statgraphics Centurion application (version 1.0.1.C)(Virginia, USA). The significance of the results was determined with an unpaired *t*–test or one–way ANOVA with Tukey’s test. A multivariate statistical data analysis (MVA) of the samples was performed with SIMCA P+ version 15 (Umetrics AB, Umeå, Sweden), after mean centering all of the variables and scaling unit-variance. The metabolic variables affected by ABA and CHT treatments were revealed through the principal component analysis (PCA), applied as the unsupervised MVA method. The heatmaps were created using the Perseus software (version 1.5.3.2) and the 2^−ΔCt^ values from gene expression analysis.

### 4.6. Meteorological Data

The climatic data were pooled from the National Observatory of Athens’ Automatic Network [87].

## 5. Conclusions

Biostimulant application is a useful viticultural practice to improve the grape and wine quality, especially in the challenging era of climate change. However, the optimization of the application time and the concentration of the compound applied are critical parameters for the outcome of the approach. In the present study, abscisic acid and chitosan were found to induce the expression of genes involved in anthocyanins and stilbenoids’ biosynthesis, and to enhance their accumulation in a white-color cultivar under the Greek vineyard conditions. Further alterations in the other phenylpropanoid gene expression profiles and the phenolic compound concentrations were also observed, but they were mostly dependent on the vintage. More extensive and in-depth research studies would be necessary in order to elucidate the grapevine responses to various biostimulants and their specific impact on each cultivar metabolic profile. The acquired knowledge, combined with the grape response mechanisms to the environmental conditions during ripening, will be beneficial to viticulturists and winemakers in order to better exploit the distinguishing quality characteristics of a certain cultivar.

## Figures and Tables

**Figure 1 plants-11-01648-f001:**
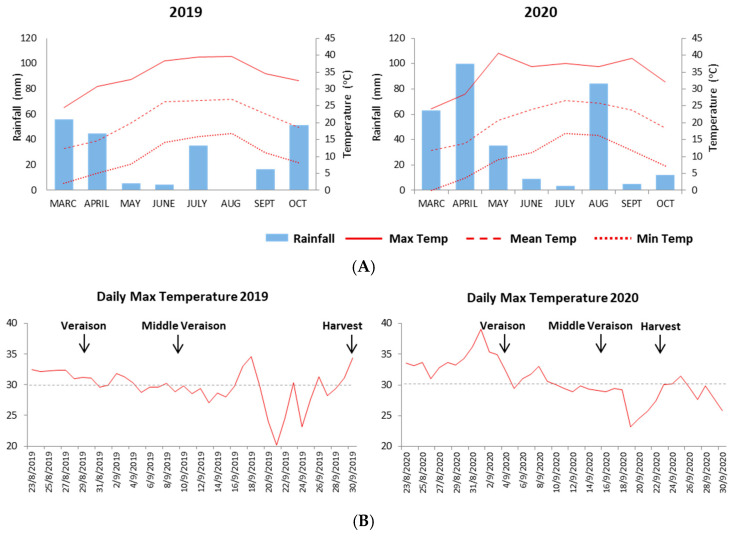
(**A**) Monthly evolution of temperatures and rainfall recorded in the Muses Valley during 2019 and 2020; (**B**) Daily evolution of maximum temperatures (above 30 °C) recorded in the Muses Valley during 2019 and 2020. Arrows indicate the three sampling dates (veraison, middle veraison and harvest).

**Figure 2 plants-11-01648-f002:**
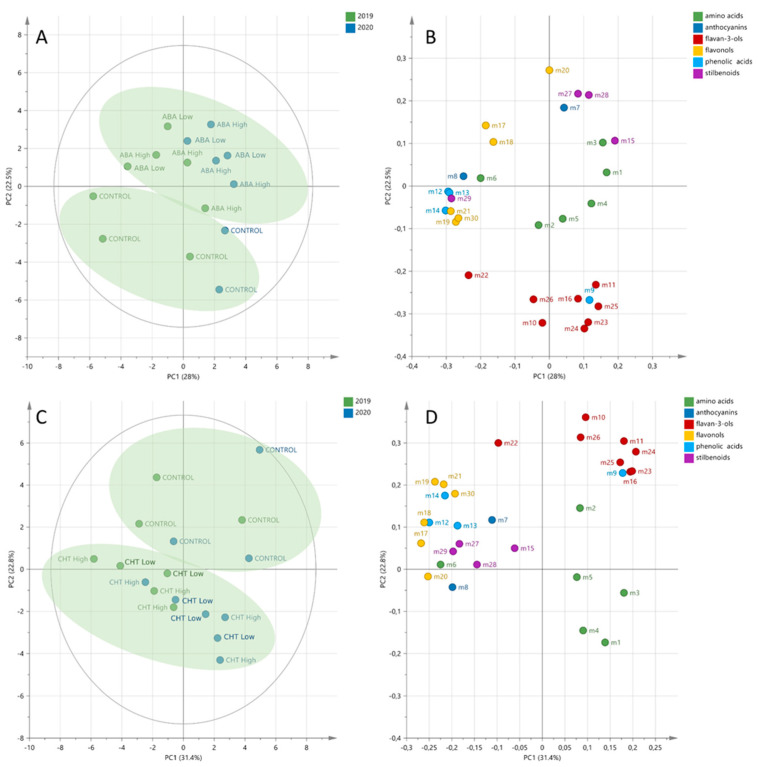
Unsupervised classification using principal component analysis on metabolomic data from grape berries of cultivar Savvatiano at veraison stage in 2019 and 2020 treated with abscisic acid (**A**,**B**) and chitosan (**C**,**D**). Samples in the score plots (**A**,**C**) were colored according to the vintage, and variables in loading plots (**B**,**D**) were colored according to the metabolic class. Numbers indicate the ID of metabolites, as follows: L-proline (m1); L-leucine (m2); L-isoleucine (m3); L-phenylalanine (m4); L-tyrosine (m5); L-tryptophan (m6); cyanidin-3-*O*-galactoside (m7); peonidin-3-*O*-(6-*p*-coumaroyl-glucoside) (m8); gallic acid (m9); catechin (m10); epicatechin (m11); coutaric acid (m12); caftaric acid (m13); fertaric acid (m14); E-piceid (m15); catechin-gallate (m16); kaempferol-3-*O*-glucoside (m17); quercetin-3-*O*-glucoside (m18); quercetin-3-*O*-glucuroside (m19); quercetin-3-*O*-glucuronide (m20); myricetin-glucoside (m21); procyanidin B1 (m22); procyanidin B2 (m23); procyanidin B3 (m24); procyanidin B4 (m25); procyanidin-gallate (m26); *E*-resveratrol (m27); *E*-piceatannol (m28); *E*-ε-viniferin (m29); kaempferol-3-*O*-rutinoside (m30).

**Figure 3 plants-11-01648-f003:**
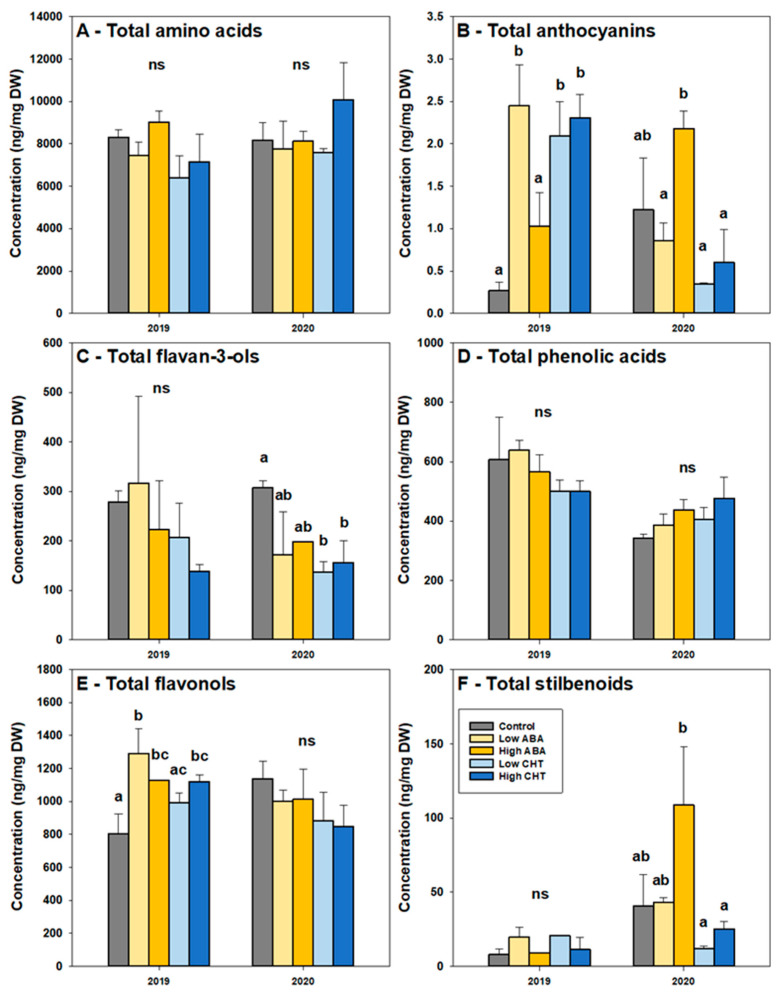
Total concentrations of amino acids (**A**), anthocyanins (**B**), flavan-3-ols (**C**), phenolic acids (**D**), flavonols (**E**) and stilbenoids (**F**) in Savvatiano berries at veraison stage in 2019 and 2020 treated with abscisic acid and chitosan: control (grey), low concentration of abscisic acid (light yellow), high concentration of abscisic acid (dark yellow), low concentration of chitosan (light blue) and high concentration of chitosan (dark blue). Error bars represent the standard deviations. No significant difference (ns) was found between values with the same letters (one-way ANOVA, *p*-value > 0.05).

**Figure 4 plants-11-01648-f004:**
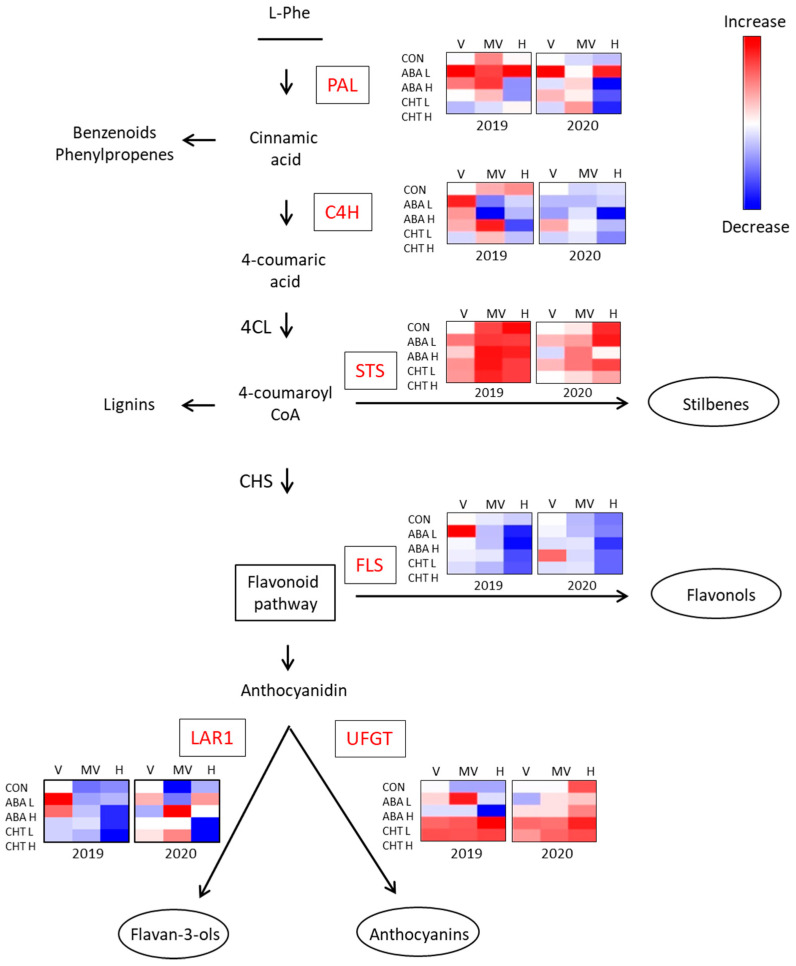
Phenylpropanoid transcriptional changes in berries treated with ABA and CHT. Variations in expression levels are shown for each treatment through a gradient color scale (blue, lower; red, higher) in comparison to the expression level in non-treated plants at veraison stage (white). V, veraison; M, middle veraison; H, harvest; CON, control; L, lower biostimulant dose; H, higher biostimulant dose.

**Table 1 plants-11-01648-t001:** Physiochemical characteristics of grapes during maturation.

2019					
Stage	Treatment	Weight/Berry (g)	Total Soluble Solids (°Brix)	pH	Total Acidity (Tart. Ac. g/L)
Veraison	Control	2.22 ± 0.29 b	14.26 ± 0.75 b	2.94 ± 0.22 a	5.42 ± 0.24 b
	ABA Low	2.50 ± 0.17 ab	15.16 ± 0.40 a	2.74 ± 0.01 a	5.40 ± 0.07 b
	ABA High	2.38 ± 0.15 ab	15.06 ± 0.40 ab	2.91 ± 0.15 a	5.15 ± 0.15 b
	CHT Low	2.62 ± 0.08 a	15.33 ± 0.37 a	2.89 ± 0.05 a	5.32 ± 0.11 b
	CHT High	2.62 ± 0.14 a	15.60 ± 0.02 a	2.78 ± 0.02 a	5.82 ± 0.18 a
Mid Veraison	Control	2.70 ± 0.16 a	17.13 ± 0.55 a	3.14 ± 0.07 ab	4.75 ± 0.11 a
	ABA Low	2.72 ± 0.11 a	17.33 ± 0.20 a	3.22 ± 0.06 ab	4.32 ± 0.11 c
	ABA High	2.32 ± 0.14 b	17.63 ± 0.77 a	3.26 ± 0.11 a	4.60 ± 0.08 ab
	CHT Low	2.44 ± 0.06 b	17.20 ± 0.51 a	3.11 ± 0.01 b	4.77 ± 0.04 a
	CHT High	2.07 ± 0.14 c	17.80 ± 0.30 a	3.21 ± 0.04 ab	4.55 ± 0.11 b
Harvest	Control	2.30 ± 0.06 ab	17.13 ± 0.55 a	3.14 ± 0.07 ab	4.75 ± 0.11 a
	ABA Low	2.55 ± 0.30 a	17.33 ± 0.20 a	3.22 ± 0.06 ab	4.32 ± 0.11 c
	ABA High	2.08 ± 0.13 b	17.63 ± 0.77 a	3.26 ± 0.11 a	4.60 ± 0.08 ab
	CHT Low	2.55 ± 0.19 a	17.20 ± 0.51 a	3.11 ± 0.01 b	4.77 ± 0.04 a
	CHT High	2.05 ± 0.17 b	17.80 ± 0.30 a	3.21 ± 0.04 ab	4.55 ± 0.11 b
**2020**					
Veraison	Control	2.85 ± 0.11 a	17.76 ± 0.28 a	3.32 ± 0.09 ab	5.30 ± 0.34 ab
	ABA Low	2.74 ± 0.02 a	17.16 ± 0.50 ab	3.21 ± 0.04 b	5.55 ± 0.15 ab
	ABA High	2.81 ± 0.11 a	17.56 ± 0.83 ab	3.73 ± 0.10 b	5.15 ± 0.22 b
	CHT Low	2.74 ± 0.26 a	17.26 ± 0.41 ab	3.29 ± 0.03 ab	5.15 ± 0.17 b
	CHT High	2.68 ± 0.08 a	16.7 ± 0.55 b	3.27 ± 0.04 ab	5.70 ± 0.15 a
Mid Veraison	Control	2.47 ± 0.13 a	19.33 ± 0.21 a	3.36 ± 0.02 ab	3.7 ± 0.1 b
	ABA Low	2.90 ± 0.04 a	18.13 ± 0.46 c	3.40 ± 0.02 a	3.70 ± 0.0 ba
	ABA High	2.72 ± 0.07 a	18.63 ± 0.55 bc	3.36 ± 0.05 ab	3.90 ± 0.19 ab
	CHT Low	2.61 ± 0.43 a	19.06 ± 0.21 ab	3.32 ± 0.02 b	3.7b ± 0.19 ab
	CHT High	2.77 ± 0.22 a	18.81 ± 0.34a bc	3.36 ± 0.03 ab	4.17 ± 0.04 a
Harvest	Control	2.94 ± 0.22 bc	20.33 ± 0.20 a	3.42 ± 0.03 a	4.35 ± 0.15 d
	ABA Low	3.08 ± 0.05 ab	18.03 ± 0.45 b	3.21 ± 0.07 c	4.50 ± 0.08 cd
	ABA High	3.26 ± 0.14 a	19.23 ± 0.55 ab	3.23 ± 0.07 c	4.85 ± 0.01 cd
	CHT Low	2.80 ± 0.14 c	19.40 ± 1.57 ab	3.36 ± 0.08 b	4.76 ± 0.0 cb
	CHT High	3.13 ± 0.11 ab	19.16 ± 1.07 ab	3.35 ± 0.03 b	5.01 ± 0.15 a

Data represent means ± SD. Different letters in the same column and phenological stage indicate significant differences according to one-way ANOVA, *p*-value > 0.05.

**Table 3 plants-11-01648-t003:** List of primers used in RT-qPCR analysis.

Gene Name	NCBI Accession Number	Forward Primer	Reverse Primer	Reference
VviPAL	XM_010660093.2	GTGAGGGAAGAACTGGGAGC	TTGTCACACTCTTCACCGGG	[86]
VviC4H	XM_002266202.3	GAACCACCTGAACCTCTCCG	ATCCGAACTCCACTCCCTGA	[86]
VviSTS	X_76892	ATCGAAGATCACCCACCTTG	CTTAGCGGTTCGAAGGACAG	[6]
VviFLS	XM_002285803.4	TGGGGTTAGGTCTGGGAGAG	AACCTGCAAGCCCTGAACTT	[61]
VviUFGT	NM_001397857.1	TGGTGGCTGACGCATTCAT	CCCCATCTCTGCTGCCATATC	[13]
VviLAR1	NM_001280958.1	CAGGAGGCTATGGAGAAGATAC	ACGCTTCTCTCTGTACATGTTG	[61]
VviEF1a	XM_002284888.3	GAACTGGGTGCTTGATAGGC	AACCAAAATATCCGGAGTAAAAGA	[86]

## Data Availability

The data presented in this study are available on request from the corresponding authors (pending privacy and ethical considerations).

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
