# Peer review of "Abscisic Acid and Chitosan Modulate Polyphenol Metabolism and Berry Qualities in the Domestic White-Colored Cultivar Savvatiano"

_plants, 2022, doi:10.3390/plants11131648_

Round 1
Reviewer 1 Report
1. The researches and the experimental scheme were clearly defined, organized, tracked and evaluated. The experimental area is very well characterized.
The grapes were collected for analysis from vines treated with two biostimulants (abscisic acid and chitosan) in three phenological stages (harvesting, middle sampling and harvesting).
The characterization methods used are relevant to the purpose proposed in the article.
2. The analysis methods, the implemented mathematical models and the statistical analysis of the experimental data were well chosen, clearly and reasonably defined and relevant for the proposed purpose, respectively: biostimulants applied (abscisic acid and chitosan), plant material used, growth conditions, climate conditions, berry ripening and conventional must analysis (berry size, total soluble solids, pH and total acidity), metabolic profile in grape berries determined with UPLC-MS (comprising of six amino acids, four phenolic acids, four stilbenoids, six flavonols, eight flavan-3-ols and two anthocyanins), gene expression examined by targeted RT-qPCR analysis.
The methods of analysis and the equipment used are relevantly described. The following were evaluated and analyzed:
- physiochemical characteristics of grapes during maturation;
- total concentrations of amino acids, anthocyanins, flavan-3-ols, phenolic acids, flavonol and stilbenoids in Savvatiano berries in 2019 and 2020, in three phenological stages;
- expression level of genes involved in phenylpropanoid pathway (VviPAL (A), VviC4H and VviUFGT in Savvatiano during two vegetative seasons (2019 and 2020), in three phenological stages.
3. The conclusions are well argued and supported by the results obtained, as well as those presented in 4 tables (supplementary data - 3 tabeles) 8 figures and the supplementary materials.
The use of biostimulants in agriculture is an innovative method of dealing with environmental stressors affecting plant growth and development.
4. References (79 articles and 13 articles in supplementary data), selected from 1995-2022 are relevant for the subject of the article.
Author Response
Dear Mr/ Mrs
On behalf of my colleagues and co-authors we would like to thank you for taking time in order to make the review of our manuscript. Thank you for providing us with an opportunity to re-read this manuscript. As noted this is a first time of the investigation of the impact of two biostimulants - abscisic acid and chitosan on polyphenol metabolism of the Greek grapevine cultivar Savvatiano. Therefore, it is an interesting topic which is gaining interest from both researcher and the public alike.
What concerns the revision we did, are listed above
In the abstract we added a sentence in order to provide to the readers the clear purpose of our work.
- Herein, we investigated the impact of two biostimulants - abscisic acid (0.04% w/v and 0.08% w/v) and chitosan (0.3% w/v and 0.6% w/v) on polyphenol metabolism of the Greek grapevine cultivar Savvatiano
Changed to
Herein, we investigated the impact of two biostimulants - abscisic acid (0.04% w/v and 0.08% w/v) and chitosan (0.3% w/v and 0.6% w/v) on polyphenol metabolism of the Greek grapevine cultivar Savvatiano, in order to determine the impact of biostimulants application in the concentration of phenolic compounds.
- At the section of the results
The experimental vineyard is located in a narrow valley on a mountain slope at approximately 450 m altitude, in the area of Central Greece, 100 Km north-west of Athens
Change to
The experimental vineyard is located in a narrow valley at approximately 450 m altitude, in the area of Central Greece, 100 Km north-west of Athens
- In the paragraph 3.3
comprising of six amino acids, four phenolic acids, four stilbenoids, six flavonols, eight flavan-3-ols and two anthocyanins
Changed to
comprising of six amino acids, four phenolic acids, four stilbenoids, six flavonols, eight flavan-3-ols and two anthocyanins di OH
- The legend of the figure 4
Phenypropanoid transcriptional changes in berries treated with treated with ABA and CHT. Variations in expression levels are shown for each treatment through a gradient color scale (blue, lower; red, higher) in comparison to the expression level in non-treated plants at veraison stage (white). V, veraison; M, middle veraison; H, harvest; CON, control; L, lowest lower biostimulant dose,; H, highest higher biostimulant dose.
Changed to
Phenypropanoid transcriptional changes in berries treated with ABA and CHT. Variations in expression levels are shown for each treatment through a gradient color scale (blue, lower; red, higher) in comparison to the expression level in non-treated plants at veraison stage (white). V, veraison; M, middle veraison; H, harvest; CON, control; L, lower biostimulant dose; H, higher biostimulant dose.
- In the first paragraph of the conclusions
The different weather conditions among the two years at the location of the vineyard influenced the maturation process, and therefore the vintage effect was evident in the grape berry physiochemical characteristics during the harvest stage
Changed to
The different weather conditions among the two years at the location of the vineyard influenced the maturation process, therefore the vintage effect was evident in the grape berry physiochemical characteristics during the harvest stage
- Moreover in the section of Conlusions, a sentence was erased due to the fact that the same similar point about the anthocyanins is being mentioned in the previous paragraph.
Although Savvatiano is a white grape variety, is the first time that small amounts of anthocyanins were detected at all treatments. Anthocyanins is well known that are related with red wine color and at the last years trace amount have been detected at several white grape berries and wines [58,59]. Although still controversy it is believed that the present of the anthocyanins in white wines could provoke the pinking phenomenon [65].
Changed to
Although Savvatiano is a white grape variety, is the first time that small amounts of anthocyanins were detected at all treatments. Although still controversy it is believed that the present of the anthocyanins in white wines could provoke the pinking phenomenon [65].
- at the section of the Supplementary data we erase the full description of the legend
The following supporting information can be downloaded at: www.mdpi.com/xxx/s1, Table S1: Date of biostimulant applications during 2019 and 2020 vintages; Table S2: List of compounds identified in this study based on MS and UV spectra. RT retention time, * tentative assignments based on MS data, UV spectra, elution order available from literature; Table S3: List of primers used in RT-qPCR analysis; Figure S1: Map of Central Greece showing the location of the experimental vineyard; Figure S2: Single vineyard in the Muses Valley (A) and the Experimental Design (B); Figure S3: Unsupersvised classification using principal component analysis on metabolomic data from grape berries of cultivar Savvatiano at middle veraisson (A, B) and harvest (C, D) stages in 2019 and 2020 treated with ABA. Samples in the score plots (A, C) were colored according to the vintage, and variables in loading plots (B, D) were colored according to the metabolic class. Numbers indicate the ID of metabolites, as follows: L-proline (m1), L-leucine (m2), L-isoleucine (m3), L-phenylalanine (m4), L-tyrosine (m5), L-tryptophan (m6), cyanidin-3-O-galactoside (m7), peonidin-3-O-(6-p-coumaroyl-glucoside) (m8), gallic acid (m9), catechin (m10), epicatechin (m11), coutaric acid (m12), caftaric acid (m13), fertaric acid (m14), E-piceid (m15), catechin-gallate (m16), kaempferol-3-O-glucoside (m17), quercetin-3-O-glucoside (m18), quercetin-O-glucuronide (m19), quercetin-3-O-glucuronide (m20), myricetin-glucoside (m21), procyanidin B1 (m22), procyanidin B2 (m23), procyanidin B3 (m24), procyanidin B4 (m25), procyanidin-gallate (m26), E-resveratrol (m27), E-piceatannol (m28), E--viniferin (m29), kaempferol-3-O-rutinoside (m30); Figure S4: Unsupersvised classification using principal component analysis on metabolomic data from grape berries of cultivar Savvatiano at middle veraison (A, B) and harvest (C, D) stages in 2019 and 2020 treated with chitosan. Samples in the score plots (A, C) were colored according to the vintage, and variables in loading plots (B, D) were colored according to the metabolic class. Numbers indicate the ID of metabolites, as follows: L-proline (m1), L-leucine (m2), L-isoleucine (m3), L-phenylalanine (m4), L-tyrosine (m5), L-tryptophan (m6), cyanidin-3-O-galactoside (m7), pe-onidin-3-O-(6-p-coumaroyl-glucoside) (m8), gallic acid (m9), catechin (m10), epicatechin (m11), coutaric acid (m12), caftaric acid (m13), fertaric acid (m14), E-piceid (m15), catechin-gallate (m16), kaempferol-3-O-glucoside (m17), quercetin-3-O-glucoside (m18), quercetin-O-glucuronide (m19), quercetin-3-O-glucuronide (m20), myricetin-glucoside (m21), procyanidin B1 (m22), procyanidin B2 (m23), procyanidin B3 (m24), procyanidin B4 (m25), procyanidin-gallate (m26), E-resveratrol (m27), E-piceatannol (m28), E--viniferin (m29), kaempferol-3-O-rutinoside (m30); Figure S5: Total concentrations of amino acids (A), anthocyanins (B), flavan-3-ols (C), phenolic acids (D), flavonols (E) and stilbenoids (F) in Savvatiano berries at middle veraison stage in 2019 and 2020 depending on the treatment : control (grey), lower concentration of abscisic acid (light yellow), higher concentration of abscisic acid (dark yellow), lower concentration of chitosan (light blue) and higher concentration of chitosan (dark blue). Error bars represent the standard deviations. No significant difference (ns) were found between values with the same letters (one-way ANOVA, p-value > 0.05); Figure S6: Total concentrations of amino acids (A), anthocyanins (B), flavan-3-ols (C), phenolic acids (D), flavonols (E) and stilbenoids (F) in Savvatiano berries at harvest stage in 2019 and 2020 depending on the treatment : control (grey), lower concentration of abscisic acid (light yellow), higher concentration of abscisic acid (dark yellow), lower concentration of chitosan (light blue) and higher concentration of chitosan (dark blue). Error bars represent the standard deviations. No significant difference (ns) were found between values with the same letters (one-way ANOVA, p-value > 0.05). Figure S7: Expression level of genes involved in phenylpropanoid pathway (VviPAL (A), VviC4H (B) and VviUFGT (C)) in Savvatiano during two vegetative seasons (2019 and 2020). Vertical bars represent the standard deviation and asterisks indicate the statistically significant differences (Student’s t-test, p–value < 0.05). The three sampling points (veraison; middle veraison and harvest) are indicated under each graphs; Figure S8: Expression levels of genes involved in phenylpropanoid pathway (VviFLS (A), VviUFGT (B) and VviLAR1 (C)) in Savvatiano during two vegetative seasons (2019 and 2020). Vertical bars represent the standard deviation and asterisks indicate the statistically significant differences (Student’s t-test, p–value < 0.05). The three sampling points (veraison; middle veraison, and harvest,) are indicated under each graphs.
Changed to
Table S1: Date of biostimulant applications during two growing seasons, Table S2: List of compounds identified by UPL-MS, Table S3: List of primers used in RT-qPCR, Figure S1: Map of Central Greece showing the location of the experimental vineyard, Figure S2: Single vineyard in the Muses Valley and the Experimental Design, Figure S3: Unsupersvised classification using PCA on metabolomic data from grape berries treated with ABA, Figure S4: Unsupersvised classification using PCA on metabolomic data from grape berries treated with CHT, Figure S5: Total concentration of polyphenols in Savvatiano berries at middle veraison stage Figure S6: Total concentration of polyphenols in Savvatiano grape berries at harvest stage Figure S7 and Figure S8: Expression level of genes involved in phenylpropanoid pathway
- at the section of the Authors Contributions some spelling and adding of the authors were done
Author Contributions: “Conceptualization, D.E.M.,A.A, and A.L.; methodology, D.E.M. and N.K.; software, D.E.M and A.L.; validation, D.E.M.,A.A, N.K., A.TA. A.K., M.U. M.P.G. and A.L.; formal analysis, D.E.M..; investigation, D.E.M. and N.K. ; resources, D.E.M and A.A. X.X.; data curation, X.X.; writing—original draft preparation, D.E.M., A.A.; writing—review and editing, D.E.M.,A.A, N.K., A.A. A.K., M.U. M.P.G. A.L., P.H. and Y.K.,; visualization, D.E.M., A.A, N.K., A.A. A.K., M.U. M.P.G.and A.L X.X.; supervision, project administration, D.E.M. and A.L.; funding acquisition, A.L.; Y.K. All authors have read and agreed to the published version of the manuscript.” Please turn to the CRediT taxonomy for the term explanation.
Changed to
Author Contributions: “Conceptualization, D.E.M.,A.A, and A.L.; methodology, D.E.M. and N.K.; software, D.E.M and A.L.; validation, D.E.M.,A.A, N.K., A.T. A.K., M.U. M.P.G. and A.L.; formal analysis, D.E.M..; investigation, D.E.M. and N.K.; resources, D.E.M. and A.A.; data curation, X.X.; writing—original draft preparation, D.E.M., A.A.; writing—review and editing, D.E.M.,A.A, N.K., A.A. A.K., M.U. M.P.G. A.L., P.H. and Y.K.,; visualization, D.E.M., A.A, N.K., A.A. A.K., M.U. M.P.G. and A.L.; supervision, project administration, D.E.M. and A.L.; funding acquisition, A.L.; Y.K. All authors have read and agreed to the published version of the manuscript.” Please turn to the CRediT taxonomy for the term explanation.
Kind regards
Dimitrios Miliordos

Reviewer 2 Report
The authors report an important analysis on the effects of chitosan and ABA treatment (2 dosages) in Greek Vitis vinifera cultures.
Macroscopic aspects were evaluated such as the weight and acidity of the grapes. The samplings were performed in three different moments of the growth of the plant. Furthermore, biochemical aspects were considered such as the content of anthocyanins, phenolic compounds, flavonols etc. Finally, the gene expression levels of the enzymes expressing the metabolism of stilbenes, flavonols, anthocyanins etc. were quantified.
The most interesting aspect is that the use of these biostimulants (chitosan and ABA) was associated with the difficult climatic conditions during the sampling periods. In fact, the harsh climatic conditions (water stress above all) can negatively affect all the biochemical parameters measured. The use of ABA and chitosan biostimulants would improve the quality and healthiness of the product.
The manuscript is well written and a lot of data has been collected and discussed, apart from some typos, in my opinion, the manuscript can be accepted for publication.
Author Response
Reviewer 2
The authors report an important analysis on the effects of chitosan and ABA treatment (2 dosages) in Greek Vitis vinifera cultures.
Macroscopic aspects were evaluated such as the weight and acidity of the grapes. The samplings were performed in three different moments of the growth of the plant. Furthermore, biochemical aspects were considered such as the content of anthocyanins, phenolic compounds, flavonols etc. Finally, the gene expression levels of the enzymes expressing the metabolism of stilbenes, flavonols, anthocyanins etc. were quantified.
The most interesting aspect is that the use of these biostimulants (chitosan and ABA) was associated with the difficult climatic conditions during the sampling periods. In fact, the harsh climatic conditions (water stress above all) can negatively affect all the biochemical parameters measured. The use of ABA and chitosan biostimulants would improve the quality and healthiness of the product.
The manuscript is well written and a lot of data has been collected and discussed, apart from some typos, in my opinion, the manuscript can be accepted for publication.
Dear Mr/ Mrs
On behalf of my colleagues and co-authors we would like to thank you for taking time in order to make the review of our manuscript. Thank you for providing us with an opportunity to re-read this manuscript. As noted this is a first time of the investigation of the impact of two biostimulants - abscisic acid and chitosan on polyphenol metabolism of the Greek grapevine cultivar Savvatiano. Therefore, it is an interesting topic which is gaining interest from both researcher and the public alike.
What concerns the revision we did, are listed above
In the abstract we added a sentence in order to provide to the readers the clear purpose of our work.
- Herein, we investigated the impact of two biostimulants - abscisic acid (0.04% w/v and 0.08% w/v) and chitosan (0.3% w/v and 0.6% w/v) on polyphenol metabolism of the Greek grapevine cultivar Savvatiano
Changed to
Herein, we investigated the impact of two biostimulants - abscisic acid (0.04% w/v and 0.08% w/v) and chitosan (0.3% w/v and 0.6% w/v) on polyphenol metabolism of the Greek grapevine cultivar Savvatiano, in order to determine the impact of biostimulants application in the concentration of phenolic compounds.
- At the section of the results
The experimental vineyard is located in a narrow valley on a mountain slope at approximately 450 m altitude, in the area of Central Greece, 100 Km north-west of Athens
Change to
The experimental vineyard is located in a narrow valley at approximately 450 m altitude, in the area of Central Greece, 100 Km north-west of Athens
- In the paragraph 3.3
comprising of six amino acids, four phenolic acids, four stilbenoids, six flavonols, eight flavan-3-ols and two anthocyanins
Changed to
comprising of six amino acids, four phenolic acids, four stilbenoids, six flavonols, eight flavan-3-ols and two anthocyanins di OH
- The legend of the figure 4
Phenypropanoid transcriptional changes in berries treated with treated with ABA and CHT. Variations in expression levels are shown for each treatment through a gradient color scale (blue, lower; red, higher) in comparison to the expression level in non-treated plants at veraison stage (white). V, veraison; M, middle veraison; H, harvest; CON, control; L, lowest lower biostimulant dose,; H, highest higher biostimulant dose.
Changed to
Phenypropanoid transcriptional changes in berries treated with ABA and CHT. Variations in expression levels are shown for each treatment through a gradient color scale (blue, lower; red, higher) in comparison to the expression level in non-treated plants at veraison stage (white). V, veraison; M, middle veraison; H, harvest; CON, control; L, lower biostimulant dose; H, higher biostimulant dose.
- In the first paragraph of the conclusions
The different weather conditions among the two years at the location of the vineyard influenced the maturation process, and therefore the vintage effect was evident in the grape berry physiochemical characteristics during the harvest stage
Changed to
The different weather conditions among the two years at the location of the vineyard influenced the maturation process, therefore the vintage effect was evident in the grape berry physiochemical characteristics during the harvest stage
- Moreover in the section of Conlusions, a sentence was erased due to the fact that the same similar point about the anthocyanins is being mentioned in the previous paragraph.
Although Savvatiano is a white grape variety, is the first time that small amounts of anthocyanins were detected at all treatments. Anthocyanins is well known that are related with red wine color and at the last years trace amount have been detected at several white grape berries and wines [58,59]. Although still controversy it is believed that the present of the anthocyanins in white wines could provoke the pinking phenomenon [65].
Changed to
Although Savvatiano is a white grape variety, is the first time that small amounts of anthocyanins were detected at all treatments. Although still controversy it is believed that the present of the anthocyanins in white wines could provoke the pinking phenomenon [65].
- at the section of the Supplementary data we erase the full description of the legend
The following supporting information can be downloaded at: www.mdpi.com/xxx/s1, Table S1: Date of biostimulant applications during 2019 and 2020 vintages; Table S2: List of compounds identified in this study based on MS and UV spectra. RT retention time, * tentative assignments based on MS data, UV spectra, elution order available from literature; Table S3: List of primers used in RT-qPCR analysis; Figure S1: Map of Central Greece showing the location of the experimental vineyard; Figure S2: Single vineyard in the Muses Valley (A) and the Experimental Design (B); Figure S3: Unsupersvised classification using principal component analysis on metabolomic data from grape berries of cultivar Savvatiano at middle veraisson (A, B) and harvest (C, D) stages in 2019 and 2020 treated with ABA. Samples in the score plots (A, C) were colored according to the vintage, and variables in loading plots (B, D) were colored according to the metabolic class. Numbers indicate the ID of metabolites, as follows: L-proline (m1), L-leucine (m2), L-isoleucine (m3), L-phenylalanine (m4), L-tyrosine (m5), L-tryptophan (m6), cyanidin-3-O-galactoside (m7), peonidin-3-O-(6-p-coumaroyl-glucoside) (m8), gallic acid (m9), catechin (m10), epicatechin (m11), coutaric acid (m12), caftaric acid (m13), fertaric acid (m14), E-piceid (m15), catechin-gallate (m16), kaempferol-3-O-glucoside (m17), quercetin-3-O-glucoside (m18), quercetin-O-glucuronide (m19), quercetin-3-O-glucuronide (m20), myricetin-glucoside (m21), procyanidin B1 (m22), procyanidin B2 (m23), procyanidin B3 (m24), procyanidin B4 (m25), procyanidin-gallate (m26), E-resveratrol (m27), E-piceatannol (m28), E--viniferin (m29), kaempferol-3-O-rutinoside (m30); Figure S4: Unsupersvised classification using principal component analysis on metabolomic data from grape berries of cultivar Savvatiano at middle veraison (A, B) and harvest (C, D) stages in 2019 and 2020 treated with chitosan. Samples in the score plots (A, C) were colored according to the vintage, and variables in loading plots (B, D) were colored according to the metabolic class. Numbers indicate the ID of metabolites, as follows: L-proline (m1), L-leucine (m2), L-isoleucine (m3), L-phenylalanine (m4), L-tyrosine (m5), L-tryptophan (m6), cyanidin-3-O-galactoside (m7), pe-onidin-3-O-(6-p-coumaroyl-glucoside) (m8), gallic acid (m9), catechin (m10), epicatechin (m11), coutaric acid (m12), caftaric acid (m13), fertaric acid (m14), E-piceid (m15), catechin-gallate (m16), kaempferol-3-O-glucoside (m17), quercetin-3-O-glucoside (m18), quercetin-O-glucuronide (m19), quercetin-3-O-glucuronide (m20), myricetin-glucoside (m21), procyanidin B1 (m22), procyanidin B2 (m23), procyanidin B3 (m24), procyanidin B4 (m25), procyanidin-gallate (m26), E-resveratrol (m27), E-piceatannol (m28), E--viniferin (m29), kaempferol-3-O-rutinoside (m30); Figure S5: Total concentrations of amino acids (A), anthocyanins (B), flavan-3-ols (C), phenolic acids (D), flavonols (E) and stilbenoids (F) in Savvatiano berries at middle veraison stage in 2019 and 2020 depending on the treatment : control (grey), lower concentration of abscisic acid (light yellow), higher concentration of abscisic acid (dark yellow), lower concentration of chitosan (light blue) and higher concentration of chitosan (dark blue). Error bars represent the standard deviations. No significant difference (ns) were found between values with the same letters (one-way ANOVA, p-value > 0.05); Figure S6: Total concentrations of amino acids (A), anthocyanins (B), flavan-3-ols (C), phenolic acids (D), flavonols (E) and stilbenoids (F) in Savvatiano berries at harvest stage in 2019 and 2020 depending on the treatment : control (grey), lower concentration of abscisic acid (light yellow), higher concentration of abscisic acid (dark yellow), lower concentration of chitosan (light blue) and higher concentration of chitosan (dark blue). Error bars represent the standard deviations. No significant difference (ns) were found between values with the same letters (one-way ANOVA, p-value > 0.05). Figure S7: Expression level of genes involved in phenylpropanoid pathway (VviPAL (A), VviC4H (B) and VviUFGT (C)) in Savvatiano during two vegetative seasons (2019 and 2020). Vertical bars represent the standard deviation and asterisks indicate the statistically significant differences (Student’s t-test, p–value < 0.05). The three sampling points (veraison; middle veraison and harvest) are indicated under each graphs; Figure S8: Expression levels of genes involved in phenylpropanoid pathway (VviFLS (A), VviUFGT (B) and VviLAR1 (C)) in Savvatiano during two vegetative seasons (2019 and 2020). Vertical bars represent the standard deviation and asterisks indicate the statistically significant differences (Student’s t-test, p–value < 0.05). The three sampling points (veraison; middle veraison, and harvest,) are indicated under each graphs.
Changed to
Table S1: Date of biostimulant applications during two growing seasons, Table S2: List of compounds identified by UPL-MS, Table S3: List of primers used in RT-qPCR, Figure S1: Map of Central Greece showing the location of the experimental vineyard, Figure S2: Single vineyard in the Muses Valley and the Experimental Design, Figure S3: Unsupersvised classification using PCA on metabolomic data from grape berries treated with ABA, Figure S4: Unsupersvised classification using PCA on metabolomic data from grape berries treated with CHT, Figure S5: Total concentration of polyphenols in Savvatiano berries at middle veraison stage Figure S6: Total concentration of polyphenols in Savvatiano grape berries at harvest stage Figure S7 and Figure S8: Expression level of genes involved in phenylpropanoid pathway
- at the section of the Authors Contributions some spelling and adding of the authors were done
Author Contributions: “Conceptualization, D.E.M.,A.A, and A.L.; methodology, D.E.M. and N.K.; software, D.E.M and A.L.; validation, D.E.M.,A.A, N.K., A.TA. A.K., M.U. M.P.G. and A.L.; formal analysis, D.E.M..; investigation, D.E.M. and N.K. ; resources, D.E.M and A.A. X.X.; data curation, X.X.; writing—original draft preparation, D.E.M., A.A.; writing—review and editing, D.E.M.,A.A, N.K., A.A. A.K., M.U. M.P.G. A.L., P.H. and Y.K.,; visualization, D.E.M., A.A, N.K., A.A. A.K., M.U. M.P.G.and A.L X.X.; supervision, project administration, D.E.M. and A.L.; funding acquisition, A.L.; Y.K. All authors have read and agreed to the published version of the manuscript.” Please turn to the CRediT taxonomy for the term explanation.
Changed to
Author Contributions: “Conceptualization, D.E.M.,A.A, and A.L.; methodology, D.E.M. and N.K.; software, D.E.M and A.L.; validation, D.E.M.,A.A, N.K., A.T. A.K., M.U. M.P.G. and A.L.; formal analysis, D.E.M..; investigation, D.E.M. and N.K.; resources, D.E.M. and A.A.; data curation, X.X.; writing—original draft preparation, D.E.M., A.A.; writing—review and editing, D.E.M.,A.A, N.K., A.A. A.K., M.U. M.P.G. A.L., P.H. and Y.K.,; visualization, D.E.M., A.A, N.K., A.A. A.K., M.U. M.P.G. and A.L.; supervision, project administration, D.E.M. and A.L.; funding acquisition, A.L.; Y.K. All authors have read and agreed to the published version of the manuscript.” Please turn to the CRediT taxonomy for the term explanation.
Kind regards
Dimitrios Miliordos

Reviewer 3 Report
Revision of the Manuscript ID: plants-1781367 entitled “Abscisic acid and chitosan modulate polyphenol metabolism and berry qualities in the domestic white-colored cultivar Savvatiano”
The manuscript investigated the impact of two biostimulators - abscisic acid (0.04% w / v and 0.08% w / v) and chitosan (0.3% w / v and 0.6% w / v) on the metabolism of the variety polyphenols. Greek vine Savvatiano. In the present study, abscisic acid and chitosan were found to induce the expression of genes involved in anthocyanins and stilbenoids biosynthesis and to enhance their accumulation in a white-color cultivar under the Greek vineyard conditions.
However, the manuscript has small weaknesses, and I have some concerns prior accepting this manuscript to the Plants. The authors must address the comments and or justify some putative limitations. In detail bellow.
The purpose of this paper is missing in the abstract.
The introduction provides an overview of data from existing literature.
The results and discussions are clearly detailed and well organized. They can be easily tracked. Statistical analysis seems to be well done.
The Materials and Methods section is described accordingly.
I recommend reviewing some minor English language mistakes.
Author Response
Dear Mr/ Mrs,
On behalf of my colleagues and co-authors we would like to thank you for taking time in order to make the review of our manuscript. Thank you for providing us with an opportunity to re-read this manuscript. As noted this is a first time of the investigation of the impact of two biostimulants - abscisic acid and chitosan on polyphenol metabolism of the Greek grapevine cultivar Savvatiano. Therefore, it is an interesting topic which is gaining interest from both researcher and the public alike.
Therefore, your comments are valuable in order our manuscript be improved.
- The purpose of this paper is missing in the abstract.
Herein, we investigated the impact of two biostimulants - abscisic acid (0.04% w/v and 0.08% w/v) and chitosan (0.3% w/v and 0.6% w/v) on polyphenol metabolism of the Greek grapevine cultivar Savvatiano
Changed to
Herein, we investigated the impact of two biostimulants - abscisic acid (0.04% w/v and 0.08% w/v) and chitosan (0.3% w/v and 0.6% w/v) on polyphenol metabolism of the Greek grapevine cultivar Savvatiano, in order to determine the impact of biostimulants application in the concentration of phenolic compounds.
- The introduction provides an overview of data from existing literature.
On behalf of our colleagues and co-authors we would like to thank you for the comment
- The results and discussions are clearly detailed and well organized. They can be easily tracked. Statistical analysis seems to be well done
On behalf of our colleagues and co-authors we would like to thank you for the comment
- The Materials and Methods section is described accordingly
On behalf of our colleagues and co-authors we would like to thank you for the comment
- I recommend reviewing some minor English language mistakes.
Regarding the revisions and mistakes are listed below
- At the section of the results
The experimental vineyard is located in a narrow valley on a mountain slope at approximately 450 m altitude, in the area of Central Greece, 100 Km north-west of Athens
Change to
The experimental vineyard is located in a narrow valley at approximately 450 m altitude, in the area of Central Greece, 100 Km north-west of Athens
- In the paragraph 3.3
comprising of six amino acids, four phenolic acids, four stilbenoids, six flavonols, eight flavan-3-ols and two anthocyanins
Changed to
comprising of six amino acids, four phenolic acids, four stilbenoids, six flavonols, eight flavan-3-ols and two anthocyanins di OH
- The legend of the figure 4
Phenypropanoid transcriptional changes in berries treated with treated with ABA and CHT. Variations in expression levels are shown for each treatment through a gradient color scale (blue, lower; red, higher) in comparison to the expression level in non-treated plants at veraison stage (white). V, veraison; M, middle veraison; H, harvest; CON, control; L, lowest lower biostimulant dose,; H, highest higher biostimulant dose.
Changed to
Phenypropanoid transcriptional changes in berries treated with ABA and CHT. Variations in expression levels are shown for each treatment through a gradient color scale (blue, lower; red, higher) in comparison to the expression level in non-treated plants at veraison stage (white). V, veraison; M, middle veraison; H, harvest; CON, control; L, lower biostimulant dose; H, higher biostimulant dose.
- In the first paragraph of the conclusions
The different weather conditions among the two years at the location of the vineyard influenced the maturation process, and therefore the vintage effect was evident in the grape berry physiochemical characteristics during the harvest stage
Changed to
The different weather conditions among the two years at the location of the vineyard influenced the maturation process, therefore the vintage effect was evident in the grape berry physiochemical characteristics during the harvest stage
- Moreover in the section of Conlusions, a sentence was erased due to the fact that the same similar point about the anthocyanins is being mentioned in the previous paragraph.
Although Savvatiano is a white grape variety, is the first time that small amounts of anthocyanins were detected at all treatments. Anthocyanins is well known that are related with red wine color and at the last years trace amount have been detected at several white grape berries and wines [58,59]. Although still controversy it is believed that the present of the anthocyanins in white wines could provoke the pinking phenomenon [65].
Changed to
Although Savvatiano is a white grape variety, is the first time that small amounts of anthocyanins were detected at all treatments. Although still controversy it is believed that the present of the anthocyanins in white wines could provoke the pinking phenomenon [65].
- at the section of the Supplementary data we erase the full description of the legend
The following supporting information can be downloaded at: www.mdpi.com/xxx/s1, Table S1: Date of biostimulant applications during 2019 and 2020 vintages; Table S2: List of compounds identified in this study based on MS and UV spectra. RT retention time, * tentative assignments based on MS data, UV spectra, elution order available from literature; Table S3: List of primers used in RT-qPCR analysis; Figure S1: Map of Central Greece showing the location of the experimental vineyard; Figure S2: Single vineyard in the Muses Valley (A) and the Experimental Design (B); Figure S3: Unsupersvised classification using principal component analysis on metabolomic data from grape berries of cultivar Savvatiano at middle veraisson (A, B) and harvest (C, D) stages in 2019 and 2020 treated with ABA. Samples in the score plots (A, C) were colored according to the vintage, and variables in loading plots (B, D) were colored according to the metabolic class. Numbers indicate the ID of metabolites, as follows: L-proline (m1), L-leucine (m2), L-isoleucine (m3), L-phenylalanine (m4), L-tyrosine (m5), L-tryptophan (m6), cyanidin-3-O-galactoside (m7), peonidin-3-O-(6-p-coumaroyl-glucoside) (m8), gallic acid (m9), catechin (m10), epicatechin (m11), coutaric acid (m12), caftaric acid (m13), fertaric acid (m14), E-piceid (m15), catechin-gallate (m16), kaempferol-3-O-glucoside (m17), quercetin-3-O-glucoside (m18), quercetin-O-glucuronide (m19), quercetin-3-O-glucuronide (m20), myricetin-glucoside (m21), procyanidin B1 (m22), procyanidin B2 (m23), procyanidin B3 (m24), procyanidin B4 (m25), procyanidin-gallate (m26), E-resveratrol (m27), E-piceatannol (m28), E--viniferin (m29), kaempferol-3-O-rutinoside (m30); Figure S4: Unsupersvised classification using principal component analysis on metabolomic data from grape berries of cultivar Savvatiano at middle veraison (A, B) and harvest (C, D) stages in 2019 and 2020 treated with chitosan. Samples in the score plots (A, C) were colored according to the vintage, and variables in loading plots (B, D) were colored according to the metabolic class. Numbers indicate the ID of metabolites, as follows: L-proline (m1), L-leucine (m2), L-isoleucine (m3), L-phenylalanine (m4), L-tyrosine (m5), L-tryptophan (m6), cyanidin-3-O-galactoside (m7), pe-onidin-3-O-(6-p-coumaroyl-glucoside) (m8), gallic acid (m9), catechin (m10), epicatechin (m11), coutaric acid (m12), caftaric acid (m13), fertaric acid (m14), E-piceid (m15), catechin-gallate (m16), kaempferol-3-O-glucoside (m17), quercetin-3-O-glucoside (m18), quercetin-O-glucuronide (m19), quercetin-3-O-glucuronide (m20), myricetin-glucoside (m21), procyanidin B1 (m22), procyanidin B2 (m23), procyanidin B3 (m24), procyanidin B4 (m25), procyanidin-gallate (m26), E-resveratrol (m27), E-piceatannol (m28), E--viniferin (m29), kaempferol-3-O-rutinoside (m30); Figure S5: Total concentrations of amino acids (A), anthocyanins (B), flavan-3-ols (C), phenolic acids (D), flavonols (E) and stilbenoids (F) in Savvatiano berries at middle veraison stage in 2019 and 2020 depending on the treatment : control (grey), lower concentration of abscisic acid (light yellow), higher concentration of abscisic acid (dark yellow), lower concentration of chitosan (light blue) and higher concentration of chitosan (dark blue). Error bars represent the standard deviations. No significant difference (ns) were found between values with the same letters (one-way ANOVA, p-value > 0.05); Figure S6: Total concentrations of amino acids (A), anthocyanins (B), flavan-3-ols (C), phenolic acids (D), flavonols (E) and stilbenoids (F) in Savvatiano berries at harvest stage in 2019 and 2020 depending on the treatment : control (grey), lower concentration of abscisic acid (light yellow), higher concentration of abscisic acid (dark yellow), lower concentration of chitosan (light blue) and higher concentration of chitosan (dark blue). Error bars represent the standard deviations. No significant difference (ns) were found between values with the same letters (one-way ANOVA, p-value > 0.05). Figure S7: Expression level of genes involved in phenylpropanoid pathway (VviPAL (A), VviC4H (B) and VviUFGT (C)) in Savvatiano during two vegetative seasons (2019 and 2020). Vertical bars represent the standard deviation and asterisks indicate the statistically significant differences (Student’s t-test, p–value < 0.05). The three sampling points (veraison; middle veraison and harvest) are indicated under each graphs; Figure S8: Expression levels of genes involved in phenylpropanoid pathway (VviFLS (A), VviUFGT (B) and VviLAR1 (C)) in Savvatiano during two vegetative seasons (2019 and 2020). Vertical bars represent the standard deviation and asterisks indicate the statistically significant differences (Student’s t-test, p–value < 0.05). The three sampling points (veraison; middle veraison, and harvest,) are indicated under each graphs.
Changed to
Table S1: Date of biostimulant applications during two growing seasons, Table S2: List of compounds identified by UPL-MS, Table S3: List of primers used in RT-qPCR, Figure S1: Map of Central Greece showing the location of the experimental vineyard, Figure S2: Single vineyard in the Muses Valley and the Experimental Design, Figure S3: Unsupersvised classification using PCA on metabolomic data from grape berries treated with ABA, Figure S4: Unsupersvised classification using PCA on metabolomic data from grape berries treated with CHT, Figure S5: Total concentration of polyphenols in Savvatiano berries at middle veraison stage Figure S6: Total concentration of polyphenols in Savvatiano grape berries at harvest stage Figure S7 and Figure S8: Expression level of genes involved in phenylpropanoid pathway
- at the section of the Authors Contributions some spelling and adding of the authors were done
Author Contributions: “Conceptualization, D.E.M.,A.A, and A.L.; methodology, D.E.M. and N.K.; software, D.E.M and A.L.; validation, D.E.M.,A.A, N.K., A.TA. A.K., M.U. M.P.G. and A.L.; formal analysis, D.E.M..; investigation, D.E.M. and N.K. ; resources, D.E.M and A.A. X.X.; data curation, X.X.; writing—original draft preparation, D.E.M., A.A.; writing—review and editing, D.E.M.,A.A, N.K., A.A. A.K., M.U. M.P.G. A.L., P.H. and Y.K.,; visualization, D.E.M., A.A, N.K., A.A. A.K., M.U. M.P.G.and A.L X.X.; supervision, project administration, D.E.M. and A.L.; funding acquisition, A.L.; Y.K. All authors have read and agreed to the published version of the manuscript.” Please turn to the CRediT taxonomy for the term explanation.
Changed to
Author Contributions: “Conceptualization, D.E.M.,A.A, and A.L.; methodology, D.E.M. and N.K.; software, D.E.M and A.L.; validation, D.E.M.,A.A, N.K., A.T. A.K., M.U. M.P.G. and A.L.; formal analysis, D.E.M..; investigation, D.E.M. and N.K.; resources, D.E.M. and A.A.; data curation, X.X.; writing—original draft preparation, D.E.M., A.A.; writing—review and editing, D.E.M.,A.A, N.K., A.A. A.K., M.U. M.P.G. A.L., P.H. and Y.K.,; visualization, D.E.M., A.A, N.K., A.A. A.K., M.U. M.P.G. and A.L.; supervision, project administration, D.E.M. and A.L.; funding acquisition, A.L.; Y.K. All authors have read and agreed to the published version of the manuscript.” Please turn to the CRediT taxonomy for the term explanation.
Kind regards
Dimitrios Miliordos
